# *HINT1* Gene Polymorphisms, Smoking Behaviour, and Personality Traits: A Haplotype Case-Control Study

**DOI:** 10.3390/ijms25147657

**Published:** 2024-07-12

**Authors:** Aleksandra Suchanecka, Agnieszka Boroń, Krzysztof Chmielowiec, Aleksandra Strońska-Pluta, Jolanta Masiak, Milena Lachowicz, Jolanta Chmielowiec, Grzegorz Trybek, Anna Grzywacz

**Affiliations:** 1Independent Laboratory of Behavioural Genetics and Epigenetics, Pomeranian Medical University in Szczecin, Powstańców Wielkopolskich 72 St., 70-111 Szczecin, Poland; aleksandra.suchanecka@pum.edu.pl (A.S.); aleksandra.stronska@pum.edu.pl (A.S.-P.); 2Department of Clinical and Molecular Biochemistry, Pomeranian Medical University in Szczecin, Powstańców Wielkopolskich 72 St., 70-111 Szczecin, Poland; agnieszka.boron@pum.edu.pl; 3Department of Hygiene and Epidemiology, Collegium Medicum, University of Zielona Góra, 28 Zyty St., 65-046 Zielona Góra, Poland; k.chmielowiec@inz.uz.zgora.pl (K.C.); j.chmielowiec@inz.uz.zgora.pl (J.C.); 4Second Department of Psychiatry and Psychiatric Rehabilitation, Medical University of Lublin, 1 Głuska St., 20-059 Lublin, Poland; jolanta.masiak@umlub.pl; 5Department of Psychology, Gdansk University of Physical Education and Sport, 80-336 Gdansk, Poland; milena.lachowicz@awf.gda.pl; 6Department of Oral Surgery, Pomeranian Medical University in Szczecin, Powstańców Wielkopolskich 72 St., 70-111 Szczecin, Poland; g.trybek@gmail.com

**Keywords:** *HINT1* gene, haplotype, smoking, nicotine, personality

## Abstract

The factors influencing the development and maintenance of nicotine dependence are numerous and complex. Recent studies indicate that smokers exhibit distinct genetic predispositions to nicotine dependence. We aimed to analyse (1) the association between rs2551038 and cigarette smoking, (2) the association of between the rs3864236–rs2526303–rs2551038 haplotype and cigarette smoking, and (3) the personality traits measured by the NEO Five-Factor Inventory in cigarette users and never-smokers. No significant differences were present in the frequency of rs2551038 genotypes and alleles in the studied cigarette users compared to the control group. Cigarette users, compared to the control group, had higher scores on the NEO-FFI Extraversion scale (*p* = 0.0011), and lower scores were obtained by the cigarette users for the NEO-FFI Openness (*p* = 0.0060), Agreeability (*p* ≤ 0.000), and Conscientiousness (*p* ≤ 0.000) scales. There was a significant positive Pearson’s linear correlation between the age and the Fagestrom test (r = 0.346; *p* < 0.0001) and the NEO-FFI Openness scale (r = 0.180; *p* < 0.0001) in the group of cigarette users. We observed significant linkage disequilibrium between rs2526303 and rs3864236 (D’ = 0.3581; *p* < 2.2204 × 10^−16^) and between rs2526303 and rs2551038 (D’ = 0.9993; *p* < 2.2204 × 10^−16^) in the tested sample. The sex-stratified haplotype analysis revealed that in the group of male never-smokers, the GTC haplotype was significantly more frequent than in the group of cigarette users (38% vs. 22%; *p* = 0.0039). The presented study reveals significant differences in personality trait scores between cases and controls. Moreover, the sex-stratified analysis showed significant differences in haplotype distribution. These results underscore the interplay between genetic predisposition, sex, and personality in nicotine-using individuals.

## 1. Introduction

Tobacco is the second most commonly used psychoactive substance, with over one billion smokers worldwide [1,2]. Tobacco use is regarded as a complex and multifaceted behaviour, influenced by both genetic and environmental factors. The factors affecting the development and maintenance of nicotine dependence are numerous and complex. These include the pharmacological effects of nicotine and the design of the tobacco product. Additionally, genetics, learned factors such as stimulus conditioning through frequent nicotine dosing, socioeconomic [3,4,5,6], and sociocultural factors such as family and peer tobacco use and ubiquitous tobacco marketing and retail availability play a role [7].

Recent studies indicate that smokers exhibit distinct genetic predispositions to nicotine dependence. These studies have identified significant associations between genetic variations and multiple aspects of smoking behaviour, including initiation, maintenance, and success in quitting [8,9]. A number of genes have been identified as potential contributors to smoking behaviour. These studies have focused on two broad groups of genes: (1) involved in response to nicotine, including its metabolism and receptors, and (2) involved in addiction, i.e., affecting the functioning of critical neurotransmitters (dopamine and serotonin, among others) [10,11].

Proteins bearing a conserved histidine triad (HIT) sequence, a HisXHisXHis motif (in which X is a hydrophobic amino acid), constitute a superfamily of enzymes designated as HIT proteins [12]. The histidine triad nucleotide-binding protein (HINT) is the first class of the HIT superfamily. It has been suggested that at least one HINT gene is present in all sequenced genomes. The human genome contains three independent HINT genes, which encode HINT1, HINT2, and HINT3 proteins. The gene encoding the HINT1 protein is located on chromosome 5q31.2, with a total length of 6160 base pairs (bp) consisting of three exons. The HINT1 messenger RNA (mRNA) is 782 bp in length and encodes a cytosolic protein of 126 amino acids in size with a relative molecular mass of approximately 14 kilodaltons (kDa) [3,7]. Nuclear magnetic resonance (NMR) and crystallography studies have demonstrated that HINT1 is one of the purine nucleotide-binding proteins. The protein exists in a homodimer structure, wherein each subunit possesses a purine base binding site and a ribose binding site [12,13,14]. *HINT1* is expressed in various tissues and distributed in humans and rodents’ liver, kidney, brain, and stomach [15,16]. Histidine triad nucleotide-binding protein 1 is a widely expressed nucleoside phosphoramide that is involved in a variety of biological processes, including the regulation of several CNS functions, tumour suppression, and mast cell activation via its interactions with several G protein-coupled receptors and transcription factors. Importantly, altered *HINT1* expression and mutation have been implicated in the progression of several disease states, including several neuropsychiatric disorders, peripheral neuropathy, and tumorogenesis [17]. The polymorphism rs2526303 is located in the non-coding region of the *HINT1* gene. This polymorphic site has been analysed in studies focusing on schizophrenia [15] and nicotine dependence (ND) [18]. The rs3864283 is located in the non-coding 3′UTR region of the analysed gene, and it was associated with schizophrenia in Chinese female subjects [15] and with nicotine dependence [18]. rs2551038 is located in an intron of the *HINT1* gene and is associated with ND [18] and schizophrenia [15]. Moreover, in CEU HapMap cell lines, the above-mentioned polymorphic site is a master regulator, SNP^20,^ associated with the expression of 20 genes [19].

Studies on the role of HINT1 in the central nervous system have shown that it interacts with the C-terminus of the μ-opioid receptor. When HINT1 is absent, there seems to be a decrease in receptor desensitisation and a blockage of protein kinase C-mediated μ-opioid receptor phosphorylation [20]. This can lead to an elevation in basal antinociception and morphine tolerance, along with a heightened response to the effects of amphetamine and apomorphine. Liu et al. [21] have highlighted the role of the HINT1 protein in various phases of substance dependence, including the reduction of morphine addiction and withdrawal symptoms. Moreover, studies [22,23] utilising *HINT1* knockout mice have indicated that *HINT1* may be involved in the aetiology of antidepressant and anxiety-like behaviours. Castelli et al. (2023) [24] conducted a study suggesting that prenatal exposure to tetra-hydrocannabinol could lead to difficulties in spatial cognitive processing and impair the effectors of hippocampal neuroplasticity. *HINT1* was found to play a significant role in this process. It has been observed that social isolation in mice can lead to the manifestation of schizophrenia-related behaviours, such as social withdrawal, anxiety disorders, cognitive problems, and sensorimotor gating issues. These behavioural changes are linked to modifications in certain genes, including *HINT1*. Social isolation serves as a dependable model for examining the impact of early-life stress on behaviours related to schizophrenia in mice [25]. Moreover, early-life stress might play a part in the onset of substance dependence as an inappropriate coping strategy. In research analysing the neuroanatomy of the hippocampus and its relationship with spatial learning ability in mice, HINT1 was detected within the stratum pyramidale in females [26]. This suggests its involvement in neuroplasticity, a factor that both influences and is influenced by substance abuse and dependency. The studies provide evidence supporting the potential involvement of the *HINT1* gene in plasticity, mood regulation, anxiety-like behaviour, and stress-coping mechanisms.

A study of the relationship between mutations in the *HINT1* gene and nicotine dependence found that smoking is associated with increased *HINT1* gene expression [18]. Furthermore, chronic nicotine administration in mice increased *HINT1* expression in the nucleus accumbens (NAc). However, this effect can be reversed by the administration of the nicotine antagonist, mecamylamine, after 24 h or after 72 h of nicotine withdrawal [18]. Jackson et al. [27] employed the conditioned place preference (CPP) and conditioned place aversion (CPA) reward tests to assess the impact of nicotine withdrawal on emotional and somatic symptoms in both *HINT1* knock-out (KO) and wild-type (WT) mice. Their findings demonstrated significant CPA in both groups following the withdrawal period. Nevertheless, in the case of *HINT1* KO mice, the administration of nicotine failed to elicit a notable CPP, while somatic withdrawal symptoms (such as pain hypersensitivity) were effectively mitigated. However, the study’s results regarding the association between the *HINT1* gene and smoking contradicted each other. In a recently conducted open, randomised trial of nicotine replacement therapy (NRT) involving a total of 374 nicotine-dependent smokers, the outcomes did not provide any evidence supporting a correlation between *HINT1* gene mutation and smoking cessation [28]. Furthermore, our previous studies [29,30] have indicated an association between polymorphic variants of the *HINT1* gene, smoking, and several interactions between polymorphic variants and personality traits that affect nicotine use.

Addictive behaviours, including smoking, are correlated with specific personality dimensions [31]. The Five-Factor Model of Personality (FFM) has become a widely accepted standard in psychology for classifying personality traits [32,33]. The FFM encompasses the five broad traits commonly referred to as the Big Five. These include agreeableness, openness to experience, neuroticism, and conscientiousness extraversion [34]. The FFM has been translated into several languages, and empirical studies demonstrate that the traits are indicative of fundamental individual differences across various cultural contexts [35,36]. This indicates that these five traits may have biological and evolutionary foundations [37,38]. The NEO Personality Inventory-Revised, developed by Costa and McCrae in 1995, is a widely used assessment tool designed to assess personality traits as outlined in the Five-Factor Model of personality [39]. This instrument has been translated into several languages, and its utility and efficacy have been examined in various settings, including a validation study conducted on an American Substance Use Disorder (SUD) cohort by Piedmont and Ciarrocchi (1999) [40]. These studies have demonstrated the reliability and validity of the NEO-PI-R instrument. The NEO-PI-R has demonstrated good internal consistency, stability of factor loadings, reliability over time, and content and criterion validity. Individuals suffering from substance use disorders frequently display elevated levels of neuroticism while concurrently exhibiting reduced levels of both conscientiousness and agreeableness in comparison to controls [41,42,43]. Attitude-related traits, including extraversion, novelty-seeking, and impulsivity, and avoidance-related traits, including neuroticism and harm avoidance, have been identified as the personality dimensions most frequently associated with smoking [44]. A study by Byrne et al. suggests that neuroticism is associated with smoking and that this association is independent of the stress experienced by individuals [45,46].

Nicotine use, abuse, and dependence are complex phenotypes with genetic and environmental factors involved in their development. Hence, in the present study, we aimed to analyse (1) the association between rs2551038 and cigarette smoking, (2) the association between the rs3864236–rs2526303–rs2551038 haplotype and cigarette smoking, and (3) the personality traits measured by the NEO Five-Factor Inventory in cigarette users and never-smokers.

## 2. Results

### 2.1. Hardy-Weinberg’s Equilibrium

The frequency distribution accorded with the HWE for cigarette users (χ^2^ = 1.363; *p* = 0.2430) and never-smokers (χ^2^ = 0.121; *p* = 0.727). The results are presented in Table 1.

### 2.2. Association of HINT1 rs2551038 Genotypes and Alleles with Cigarette Smoking

No significant differences were present in the frequency of rs2551038 genotypes in the studied cigarette users compared to the control group. The rs2551038 genotype distribution was C/C (0.78) vs. C/C (0.76). G/G 0.02 vs. G/G 0.02; C/G 0.20 vs. C/G 0.22, χ^2^ = 0.183, *p* = 0.9121). Additionally, no significant differences were found between the allele frequencies in the analysed groups (C 0.88 vs. C 0.87; G 0.12 vs. G 0.13, χ^2^ = 0.080, *p* = 0.7732). The results are presented in Table 2.

### 2.3. Personality Trait Assessment as Measured by the NEO Five-Factor Inventory

Cigarette users, compared to the control group, had higher scores on the NEO-FFI Extraversion scale (5.96 vs. 5.25; Z = 3.273; *p* = 0.0011). On the other hand, lower scores were obtained by the cigarette users for the NEO-FFI Openness (5.20 vs. 5.69; Z = −2.750; *p* = 0.0060), Agreeability (5.25 vs. 6.35; Z = −4.718; *p* ≤ 0.000), and Conscientiousness (5.82 vs. 6.76; Z = −4.251; *p* ≤ 0.000) scales. The results are presented in Table 3.

There was a significant positive Pearson’s linear correlation between the age and the Fagestrom scale (r = 0.346; *p* < 0.0001) and the NEO-FFI openness scale (r = 0.180; *p* < 0.0001) in the group of cigarette users. There were no significant correlations in the control group. The results are presented in Table 4.

### 2.4. Linkage Disequilibrium and Haplotype Association Analysis

We observed significant linkage disequilibrium (LD) between rs2526303 and rs3864236 (D’ = 0.3581; *p* < 2.2204 × 10^−16^) and between rs2526303 and rs2551038 (D’ = 0.9993; *p* < 2.2204 × 10^−16^) in the tested sample, including males and females (Table 5).

Sex-stratified LD analysis revealed significant disequilibrium between rs2526303 and rs3864236 (D’ = 0.7760, *p* < 2.2204 × 10^−16^); between rs2551038 and rs3864236 (D’ = 0.9985; *p* = 7.8633e × 10^−7^); and between rs2551038 and rs2526303 (D’ = 0.9993, *p* < 2.2204 × 10^−16^) in males (Table 6). In the female sample, all analysed polymorphic sites were in significant linkage disequilibrium: rs2526303 and rs3864236 (D’ = 0.7177, *p* < 2.2204 × 10^−16^), rs2551038 and rs3864236 (D’ = 0.9985; *p* = 7.8633 × 10^−7^), and lastly, rs2526303 and rs2551038 (D’ = 0.9993, *p* < 2.2204 × 10^−16^). The results are presented in Table 7.

Haplotype association analysis performed in the group of cigarette users and never-smokers revealed a significantly higher frequency of GTC haplotype in the control group compared to cigarette users, which was not significant after Bonferroni correction (27% vs. 21%; *p* = 0.0195; Table 8). The subsequent sex-stratified haplotypic analysis showed similar results in a group of males—the GTC haplotype was significantly more frequent in never-smokers than cigarette users and remained significant after the Bonferroni correction (38% vs. 22%; *p* = 0.0039, Table 9). Additionally, the ACC haplotype was significantly more frequent in a group of cigarette users than in controls, which was not significant after Bonferroni correction (57% vs. 44%; *p* = 0.0379; Table 9). A haplotype association analysis conducted on the female cohort did not identify any statistically significant differences in haplotype frequencies (Table 10).

## 3. Discussion

The present study aimed to analyse (1) the association between rs2551038 and cigarette smoking, (2) the association between the rs3864236–rs2526303–rs2551038 haplotype and cigarette smoking, and (3) the personality traits measured by the NEO Five-Factor Inventory in cigarette users and never-smokers.

The frequency of rs2551038 genotypes and alleles was found to be similar in cigarette users and never-smokers. Association analysis performed in separate groups according to sex was also non-significant. Our results regarding rs3864236 and rs2526303 have been published previously [29,30]. Before performing the haplotype association analysis, we calculated the linkage disequilibrium (LD). LD describes and measures the nonrandom association of alleles at different *loci* [47]. We observed significant LD in the tested sample between rs2526303, rs3864236 and rs2551038. Sex-stratified LD analysis revealed significant disequilibrium between all three analysed polymorphic sites, separately in groups of males and females. Haplotype association analysis performed in the group of cigarette users and never-smokers revealed a significantly higher frequency of GTC haplotype in the group of never-smokers compared to cigarette users, which was not significant after Bonferroni correction. Subsequent sex-stratified haplotype analysis showed similar results in a group of males—the GTC haplotype was significantly more frequent in never-smokers compared to cigarette users. Additionally, the ACC haplotype was significantly more frequent in a group of cigarette users than in controls; the association was not significant after Bonferroni correction. A haplotype association analysis was conducted in the female group and did not reveal significant differences.

Association and expression studies reveal the involvement of the *HINT1* gene in schizophrenia [15,48,49,50,51,52] and bipolar disorder [53,54]. In the central nervous system, the levels of HINT1 were elevated in individuals diagnosed with major depressive disorder (MDD) without psychosis [55] and reduced in individuals with schizophrenia [23]. *HINT1* KO mice lacking *HINT1* expression were found to have altered striatal and nucleus accumbens postsynaptic dopamine transmission and increased circulating corticosterone levels, which suggests that there may be effects on the hypothalamic-pituitary-adrenal axis. Interestingly, the animals showed anxiety-like behaviour [23,56]. The findings suggest that the *HINT1* gene may play a pivotal role in mood regulation. HINT1 is involved in regulating the effects of drug abuse. Studies have shown that HINT1 interacts with the C-terminus of the μ-opioid receptor, leading to attenuation of receptor desensitisation and inhibition of PKC-mediated μ-opioid receptor phosphorylation [20]. *HINT1* knockout mice are more resistant to pain than their wild-type counterparts. They show enhanced basal and morphine-induced antinociception as well as increased morphine tolerance [20]. *HINT1* knockout mice are hypersensitive to the locomotor-activating effects of amphetamine and the dopamine-receptor agonist apomorphine. This suggests that a lack of *HINT1* is associated with dysregulation of postsynaptic dopamine transmission [56]. In research conducted by Jackson et al. (2012) [57], a series of behavioural tests was used to shed light on the role of *HINT1* in acute behaviour influenced by nicotine. The study utilised both male and female *HINT1* wild-type and knockout mice. The findings revealed that male *HINT1* KO mice showed a decreased sensitivity to acute nicotine-induced antinociception in the tail-flick test, but this was not the case in the hot plate test. When exposed to low doses of nicotine, both male and female *HINT1* KO mice showed a diminished sensitivity to nicotine-induced hypomotility, with the effect being more noticeable in females. The initial variances in locomotor activity seen in male *HINT1* wild-type and KO mice were not mirrored in females. In male *HINT1* KO mice, nicotine did not trigger an anxiolytic effect; rather, it triggered an anxiogenic response. Moreover, diazepam failed to trigger an anxiolytic effect in these mice, suggesting that the observed anxiety phenotype was not reliant on nicotine. No significant variances in anxiety-like behaviour were observed in female mice.

In our previous study, there were significant differences in the frequency of both the genotypes and alleles of the rs3864283 [30] and no significant differences in the frequency of the rs2526303 [29] genotypes and alleles. In a study by Jackson (2011) [18] on analysing the three *HINT1* polymorphic sites that we describe, they found significant associations between rs2526303 and rs3864283 with nicotine dependence, the number of cigarettes smoked daily, and the Fagerström Test of nicotine dependence score. Moreover, haplotype analysis performed in this study showed that GCA and GTC (rs2551038–rs3864283–rs2526303) haplotypes were associated with the Fagerström Test of nicotine dependence score, and GCA and GTG haplotypes were associated with the number of cigarettes smoked daily. Additionally, the *HINT1* expression study revealed a trend for higher expression of *HINT1* in smokers, and together with smoking status and rs3864283, the results were significant. A randomised controlled study on smoking cessation in a nicotine replacement therapy trial showed that the *HINT1* rs3852209 TT genotype was associated with significantly higher abstinence rates at the 6-month follow-up. Still, this effect may not be related to pharmacogenetics, as participants were drug-free during this phase [28]. Interestingly, *HIINT1* effects are sex-dependent. In our allele and genotype analyses, we did not observe these effects. However, in the haplotype analysis, we observed sex-specific associations. KO mice are more sensitive to the hyperlocomotor effects of amphetamine and apomorphine compared to wild-type mice. This suggests a potential association between HINT1 deficiency and dysregulation of postsynaptic dopamine transmission [56]. It is potentially possible that such alterations in dopamine transmission may play a role in various psychoactive substance responses and are involved in basal locomotor activity differences in males but not in female mice. It has been postulated that altered dopaminergic transmission plays a role in the pathophysiology of schizophrenia [58,59]. Similar effects have been observed in individuals diagnosed with schizophrenia, with more prominent effects observed in males compared with females [15,52]. Therefore, it is important to ascertain whether *HINT1* dopamine alterations are more prevalent in males and if these alterations contribute to specific behavioural traits related to substance abuse or other psychiatric disorders.

In our study, the cigarette users, compared to the control group, had higher scores on the NEO-FFI Extraversion scale. On the other hand, lower scores were obtained by the cigarette users for the NEO-FFI Openness, Agreeability, and Conscientiousness scales. Moreover, there was a significant positive correlation between the age, the Fagestrom test, and the NEO-FFI openness scale in the group of cigarette users. Results of a study by Varadarajulu et al. [23] indicate a role for *HINT1* in modulating anxiety-related and stress-coping behaviours in mice. In our studies [29,30] regarding *HINT1* variants and personality traits, we did not observe interactions regarding anxiety as a trait or as a state, nor neuroticism, which is related to anxiety.

Aspects of addictive behaviour in general, and smoking behaviour in particular, have been associated with specific personality dimensions [31]. The participants in the study by Kulkarni et al. [60] had high levels of neuroticism, average levels of extraversion and openness, and low levels of agreeableness and conscientiousness. The personality profile of the subjects in the study by Kulkarni et al. [60] was similar to that of the smokers in the study by Terracciano et al. [61]. The study also found that, compared with non-smokers, smokers had higher scores on the neuroticism scale and lower scores on the conscientiousness scale. In our research, the conscientiousness of smokers was lower than that of the control group, but we did not observe a significant difference in neuroticism.

Avoidance-related traits, such as neuroticism, include aspects of anxiety, negative affect (e.g., depression), and anger. Individuals who score high on measures of avoidance-related characteristics may tend to smoke to alleviate high baseline levels of anxiety and negative affect after nicotine. This leads to the hypothesis of self-medication for nicotine use. According to this hypothesis, nicotine products are not used to enhance positive affect (as in classical addiction paradigms) but to alleviate negative affect through self-medication [62,63].

As previously studied by Conrod et al. [64], knowing the personality traits responsible for quitting for a particular reason and the personality traits associated with relapse will help us formulate different psychotherapeutic interventions. A study by Kulkarni et al. [60] observed that individuals with higher scores for extraversion and openness were more likely to cite health problems as a reason for quitting smoking. In contrast, those who scored higher on agreeableness were more likely to cite social factors, such as pressure from family or friends, as a reason for quitting. Individuals who scored significantly higher on conscientiousness cited social factors as a reason for quitting smoking compared to those who did not. This may be due to their greater tendency to adhere to socially defined norms. Individuals who exhibited higher levels of extraversion and agreeableness were more likely to relapse due to occupational factors such as poor performance at work, work characteristics, and frequent work-related travel. Conversely, clients who exhibited lower levels of these two personality traits were more likely to relapse due to social factors, which included peer and family tensions [60].

The results of the present study show the association between *HINT1* gene variants, personality traits, and smoking behaviour in smokers. However, we need to address its limitations. (1) Addictive behaviours are complex traits. We analysed three markers located in one gene. In the future, we plan to investigate a dense set of polymorphic sites in the *HINT1* gene and epigenetic modifications in the promoter region to enrich our analysis and understanding of *HINT1* involvement in substance dependency. (2) Our study was performed on a relatively small group of cigarette smokers of Caucasian origin. In the future, the analysis should be replicated with a larger group of participants of both sexes and other ethnicities, and more substance-dependent phenotypes should be included. (3) Among the factors relevant to nicotine usage, we considered genetics and personality. One of the environmental factors highly relevant to substance abuse is socioeconomic status, which was not analysed in the study.

Future research could take several directions to further explore and validate the interplay between the *HINT1* gene, smoking behaviour, and personality traits. (1) Conducting longitudinal research to track changes in smoking behaviour over time in individuals with different *HINT1* gene polymorphic variants, which could provide insights into the long-term effects of these genetic variants. (2) Exploring additional polymorphic sites in the *HINT1* gene or neighbouring genes that may contribute to nicotine dependence. A broader genetic analysis could uncover more complex genetic networks involved in smoking behaviour. Additionally, the analysis of *HINT1* promoter methylation could be insightful in terms of understanding its function in substance-dependent phenotypes. (3) Investigating the biological mechanisms by which the *HINT1* gene polymorphisms influence smoking behaviour. Functional studies could include cellular or animal models to understand the gene’s role in nicotine addiction. (4) Examining the associations found in our study across different cultures and populations could help determine if the observed relationships are consistent across diverse groups or if cultural factors play a significant role. (5) Testing personalised intervention strategies based on the personality profiles identified in our study could involve developing tailored smoking cessation programmes and evaluating their effectiveness. (6) Research how individuals with different *HINT1* gene polymorphic variants respond to various pharmacological treatments for smoking cessation, which could lead to more personalised medicine approaches. (7) Utilising neuroimaging techniques to observe brain activity patterns associated with the *HINT1* gene polymorphic variants in smokers and non-smokers could reveal neural correlates of the genetic predisposition to smoking.

## 4. Materials and Methods

### 4.1. Participants

The studied group comprised 522 participants: 371 cigarette users (mean age = 29.44, SD = 10.74; F = 49%, M = 51%) and 151 never-smokers (mean age = 26.91, SD = 10.10; F = 80%, M = 20%). The study was accepted by the Pomeranian Medical University in Szczecin Bioethics Committee (KB-0012/106/16). All volunteers signed written informed consent before entering the study.

The study was conducted in the Independent Laboratory of Behavioural Genetics and Epigenetics at Pomeranian Medical University in Szczecin. Both the cigarette users and controls were recruited and examined by a psychiatrist using the Mini International Neuropsychiatric Interview (MINI) and the NEO Five-Factor Personality Inventory (NEO-FFI).

### 4.2. Psychometric Tests

The Mini-International Neuropsychiatric Interview is a structured diagnostic instrument designed to evaluate diagnoses in accordance with the Diagnostic and Statistical Manual of Mental Disorders, 4th Edition (DSM-IV) and International Classification of Diseases, 10th Revision (ICD-10) criteria; it was used to exclude volunteers with neuropsychiatric disorders and substance dependencies other than nicotine.

The Five-Factor Inventory (NEO-FFI) analyses five traits—neuroticism, extroversion, openness to experience, agreeableness, and conscientiousness [65]. The results were expressed as sten scores. The conversion of the raw score to the sten scale was conducted in accordance with the Polish standards for adults, with a sten value of 1–2 corresponding to very low results, 3–4 corresponding to low results, 5–6 corresponding to average results, 7–8 corresponding to high results, and 9–10 corresponding to very high results [66].

### 4.3. Laboratory Analysis

The genomic DNA was extracted from venous blood using the QIAamp Blood DNA Mini Kit (QIAGEN, Hilden, Germany). The genotyping process was carried out using the real-time PCR technique on the LightCycler 480II instrument (Roche Diagnostics, Basel, Switzerland) with the LightSNiP (TiBMolBiol, Berlin, Germany) oligonucleotide assay. A graph was created to depict the variation in fluorescence signal in relation to temperature, thus producing a melting curve for each sample. The *HINT1* gene rs2551038 peaks were read at 53.59 °C for the C allele and 62.64 °C for the G allele.

### 4.4. Statistical Analysis

A concordance between the genotype and allele distribution and Hardy-Weinberg’s equilibrium (HWE) was tested using the HWE online software (https://wpcalc.com/en/equilibrium-hardy-weinberg (accessed on 5 April 2023)).

The Chi^2^ test was used to analyse differences in genotype and allele frequencies in both study groups.

The NEO Five-Factor Inventory trait scores (neuroticism, extraversion, openness, agreeability, and conscientiousness) were compared using the U Mann-Whitney test. The analysed variables were not distributed normally, and the homogeneity variance condition was fulfilled (Levene test *p* > 0.05).

The relationship between the age and the personality traits measured by the NEO Five-Factor Inventory and the Fagestrom test results was analysed separately in both study groups using Pearson’s linear correlation.

The computations mentioned above were performed on STATISTICA 13 software (Tibco Software Inc., Palo Alto, CA, USA) for Windows (Microsoft Corporation, Redmond, WA, USA).

Genotyping data regarding rs3864283 [30] and rs2526303 [29] were published previously and, in this study, were calculated with rs2551038 to analyse LD and haplotype frequencies in the analysed groups. Linkage disequilibrium (LD) and haplotype frequencies were estimated using the R environment version 4.3.0 (packages haplo.stats and genetics) and compared between study groups using a Chi^2^ test.

## 5. Conclusions

We found no significant differences in the frequency of rs2551038 genotypes and alleles between cigarette users and the control group. Cigarette users exhibited distinct personality profiles. They scored significantly higher on the NEO-FFI Extraversion scale, suggesting a propensity for social engagement and risk-taking. Conversely, their scores on the NEO-FFI Openness, Agreeability, and Conscientiousness scales were lower. Additionally, there was a significant positive correlation between age, the Fagestrom test, and the openness trait in the group of cigarette users. The sex-stratified analysis revealed the GTC haplotype was more frequent among male never-smokers. We did not observe this association in the female group.

The presented study reveals significant differences in personality trait scores. Moreover, the sex-stratified analysis showed significant haplotype differences in males. These results underscore the interplay between genetic predisposition, sex, and personality in nicotine-using individuals.

## Figures and Tables

**Table 1 ijms-25-07657-t001:** Hardy-Weinberg’s equilibrium for *HINT1* rs2551038 in a group of cigarette users and never-smokers.

Hardy-Weinberg Equilibrium, Including Analysis for Ascertainment Bias		Observed (Expected)	Allele Frequency	χ^2^ (*p*-Value)
Cigarette users n = 371	C/C	288 (285.6)	p (ins)= 0.88 q (del)= 0.12	1.3632 (0.2430)
	C/G	75 (79.8)		
	G/G	8 (5.6)		
Never-smokers n = 151	C/C	115 (114.5)	p (ins)= 0.87 q (del)= 0.13	0.1215 (0.7274)
	C/G	33 (34.0)		
	G/G	3 (2.5)		

*p*—statistical significance χ^2^ test.

**Table 2 ijms-25-07657-t002:** Frequency of the *HINT1* rs2551038 genotypes and alleles in the cigarette users and never-smokers groups.

	Genotypes			Alleles	
	C/C n (%)	C/G n (%)	G/G n (%)	C n (%)	G n (%)
Cigarette users n = 371	288 (77.63%)	75 (20.22%)	8 (2.16%)	651 (87.74%)	91 (12.26%)
Never-smokers n = 151	115 (76.16%)	33 (21.85%)	3 (1.99%)	263 (87.09%)	39 (12.91%)
χ^2^ (*p*-value)	0.183 (0.9121)			0.080 (0.7732)	

n—number of subjects.

**Table 3 ijms-25-07657-t003:** NEO Five-Factor Inventory sten scores in cigarette users and never-smokers.

NEO Five-Factor Inventory	Cigarette Users (n = 371) M ± SD	Never-Smokers (n = 151) M ± SD	Z	(*p*-Value)
Neuroticism scale	5.93 ± 2.22	5.71 ± 1.99	1.460	0.1442
Extraversion scale	5.96 ± 2.09	5.25 ± 1.96	3.273	0.0011 **
Openness scale	5.20 ± 2.03	5.69 ± 2.01	−2.750	0.0060 **
Agreeability scale	5.25 ± 2.25	6.35 ± 2.37	−4.718	0.0000 **
Conscientiousness scale	5.82 ± 2.13	6.76 ± 2.25	−4.251	0.0000 **

*p*-value—statistical significance; Z—Mann-Whitney U-test statistics; n—number of subjects; M ± SD, mean ± standard deviation; ** statistically significant differences after Bonferroni correction was used and *p*-value was reduced to 0.01 (*p* = 0.05/5 (number of statistical tests conducted)).

**Table 4 ijms-25-07657-t004:** Pearson’s linear correlation between age and the NEO Five-Factor Inventory traits and the Fagestrom test in cigarette users and never-smokers.

	Fagestrom Test	Neuroticism Scale	Extraversion Scale	Openness Scale	Agreeability Scale	Conscientiousness Scale
Cigarette Users (n = 371)	r = 0.346 (*p* < 0.0001) **	r = −0.025 (*p* = 0.635)	r = 0.060 (*p* = 0.248)	r = 0.180 (*p* < 0.0001) **	r = −0.071 (*p* = 0.169)	r = −0.059 (*p* = 0.257)
Never-smokers (n = 151)	-	r = −0.078 (*p* = 0.328)	r = 0.103 (*p* = 0.195)	r = −0.078 (*p* = 0.329)	r = −0.121 (*p* = 0.127)	r = −0.053 (*p* = 0.506)

r—Pearson’s linear correlation; *p*-value of statistical significance, ** statistically significant differences after Bonferroni correction was used, and *p*-value was reduced to 0.0083 (*p* = 0.05/6 (number of statistical tests conducted)).

**Table 5 ijms-25-07657-t005:** Linkage disequilibrium of rs3864236, rs2526303, and rs2551038 in the *HINT1* gene in a group of cigarette users and never-smokers.

	rs2526303	rs2551038
rs3864236 D	0.0591	−0.0003
D’	0.3581	0.0105
Chi^2^	77.4211	0.0058
*p*-value	<2.2204 × 10^−16^ **	0.9392
n	516	516
rs2526303 D		0.0742
D’		0.9993
Chi^2^		225.3464
*p*-value		<2.2204 × 10^−16^ **
n		516

D’—Levontin’s standardised disequilibrium coefficient, Chi^2^—test statistics, n—number of subjects; ** statistically significant differences after Bonferroni correction was used and *p*-value was reduced to 0.0016 (*p* = 0.05/3 (number of statistical tests conducted)).

**Table 6 ijms-25-07657-t006:** Linkage disequilibrium of rs3864236, rs2526303, and rs2551038 in the *HINT1* gene in a group of male cigarette users and never-smokers.

	rs2526303	rs2551038
rs3864236 D	0.1300	−0.0355
D’	0.7760	0.9985
Chi^2^	148.0545	24.3910
*p*-value	<2.2204 × 10^−16^ **	7.8633 × 10^−7^ **
n	215	215
rs2526303 D		0.0741
D’		0.9993
Chi^2^		89.0007
*p*-value		<2.2204 × 10^−16^ **
n		215

D’—Levontin’s standardised disequilibrium coefficient, Chi^2^—test statistics, n—number of subjects; ** statistically significant differences after Bonferroni correction was used and *p*-value was reduced to 0.0016 (*p* = 0.05/3 (number of statistical tests conducted)).

**Table 7 ijms-25-07657-t007:** Linkage disequilibrium of rs3864236, rs2526303, and rs2551038 in the *HINT1* gene in a group of female cigarette users and never-smokers.

	rs2526303	rs2551038
rs3864236 D	0.1183	−0.0309
D’	0.7177	0.9985
Chi^2^	186.5400	28.4548
*p*-value	<2.2204 × 10^−16^ **	9.5907 × 10^−8^ **
n	300	300
rs2526303 D		0.0745
D’		0.9993
Chi^2^		136.9480
*p*-value		<2.2204 × 10^−16^ **
n		300

D’—Levontin’s standardised disequilibrium coefficient, Chi^2^—test statistics, n—number of subjects, ** statistically significant differences after Bonferroni correction was used and *p*-value was reduced to 0.0016 (*p* = 0.05/3 (number of statistical tests conducted)).

**Table 8 ijms-25-07657-t008:** Haplotype association analysis for rs3864236, rs2551038, and rs2526303 in the *HINT1* gene in a group of cigarette users and never-smokers.

Haplotype	Never-Smokers n = 147	Cigarette Users n = 370	*p*-Value
G T C	0.2741	0.2060	0.0195 *
A T G	0.1190	0.1229	0.8646
G C C	0.0421	0.0466	0.7961
A T C	0.0319	0.0426	0.4619
A C C	0.5326	0.5816	0.1372

*—significant statistical differences; n—number of subjects.

**Table 9 ijms-25-07657-t009:** Haplotype association analysis for rs3864236, rs2551038, and rs2526303 in the *HINT1* gene in a group of male cigarette users and never-smokers.

Haplotype	Never-Smokers n = 30	Cigarette Users n = 186	*p*-Value
G T C	0.3822	0.2153	0.0039 **
G C C	0.0459	0.0406	0.7089
A T G	0.1114	0.1270	0.7708
A T C	0.0177	0.0481	0.2956
A C C	0.4373	0.5668	0.0379 *

*—significant statistical differences; n—number of subjects; ** statistically significant differences after Bonferroni correction was used and the *p*-value was reduced to 0.01 (*p* = 0.05/5 (number of statistical tests conducted)).

**Table 10 ijms-25-07657-t010:** Haplotype association analysis for rs3864236, rs2551038, and rs2526303 in the *HINT1* gene in a group of female cigarette users and never-smokers.

Haplotype	Never-Smokers n = 117	Cigarette Users n = 184	*p*-Value
G T C	0.2465	0.1966	0.1623
A T G	0.1196	0.1168	0.9180
A T C	0.0354	0.0370	0.9554
G C C	0.0397	0.0506	0.5447
A C C	0.5585	0.5988	0.3272

n—number of subjects.

## Data Availability

Detailed genotyping and psychometric test results are available upon request.

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
