# Peer review of "HINT1 Gene Polymorphisms, Smoking Behaviour, and Personality Traits: A Haplotype Case-Control Study"

_ijms, 2024, doi:10.3390/ijms25147657_

Round 1

Reviewer 1 Report

Comments and Suggestions for Authors 1. Authors of the manuscript (ijms-3016060) showed the impact of the HINT1 gene variants on smoking behaviour, including on the interplay of genetic variants and personality traits in smokers. They observed an effect of the rs2551038 genotype interaction and cigarette-using status on the NEO-FFI Conscientiousness scale and linkage disequilibrium between rs2526303 and rs3864236 and between rs2526303 and rs2551038 in the tested samples. Haplotype association analysis revealed a significantly higher frequency of GTC haplotype in controls than cigarette users  with similar results in males and females. The GTC haplotype was significantly more frequent in non-smokers than in smokers, whereas the ACC haplotype was significantly more frequent in a group of smokers than in non-smokers.

2. Strengths and weaknesses of the study:

Strengths:

-Research question -answered.

-Review of the literature review done correctly.

-Methods and data presented correctly.

-Analyze of the results and discussion presented correctly.

-Comparison with other studies carried out properly.

Weaknesses:

-The study group could be larger, which the authors included in the study limitations.

3. Point-by-point list of your recommendations for the improvement of the manuscript:

-It needs to be shortly described in abstract the role of HINT1 gene and its polymorphism in the body.

Author Response

Dear Reviewer,

We would like to thank you for your valuable comments on the article. Below, you will find our reply to your review. All changes are with a description or a comment, changes have been made to the manuscript (track changes in the tracking group on the review tab).

Comments and Suggestions for Authors

  1. Authors of the manuscript (ijms-3016060) showed the impact of the HINT1 gene variants on smoking behaviour, including on the interplay of genetic variants and personality traits in smokers. They observed an effect of the rs2551038 genotype interaction and cigarette-using status on the NEO-FFI Conscientiousness scale and linkage disequilibrium between rs2526303 and rs3864236 and between rs2526303 and rs2551038 in the tested samples. Haplotype association analysis revealed a significantly higher frequency of GTC haplotype in controls than cigarette users with similar results in males and females. The GTC haplotype was significantly more frequent in non-smokers than in smokers, whereas the ACC haplotype was significantly more frequent in a group of smokers than in non-smokers.

  1. Strengths and weaknesses of the study:

 Strengths:

 -Research question -answered.

 -Review of the literature review done correctly.

 -Methods and data presented correctly.

 -Analyze of the results and discussion presented correctly.

 -Comparison with other studies carried out properly.

 Weaknesses:

 -The study group could be larger, which the authors included in the study limitations.

Thank you for the comprehensive analysis of the manuscript.

  1. Point-by-point list of your recommendations for the improvement of the manuscript:

 -It needs to be shortly described in abstract the role of HINT1 gene and its polymorphism in the body.

Thank you for this suggestion. Due to the complicated nature of the HINT1 gene, more information regarding the gene and the analysed polymorphic sites were added in the introduction section, lines 81-120.

Reviewer 2 Report

Comments and Suggestions for Authors

The study “Exploring the Interplay of HINT1 Gene Polymorphisms, Smoking Behavior, and Personality Traits: A Haplotype Case-Control Study” was submitted to MDPI International Journal of Molecular Sciences.

The study is excellently written and demonstrates a methodological approach to study gene-behaviour interactions. Finding a genetic basis for smoking is relevant because, potentially, this may in the future lead into a gene therapy approach for individuals with a high dependence on nicotine or at least a better understanding of the underlying mechanisms. Understanding underlying mechanisms of addiction is therefore significant and relevant for the readers of Molecular Sciences.

On a methodological level, I would kindly like to ask for better explanation of the research questions, hypothesis and rationale. Specifically, the causal relationships between genetic, smoking status and personality trait are not clear to me. Causal relationships between a gene/allel and a certain trait are easier to demonstrate if they are very directly linked such as the CFTR gene and the transmembrane protein it codes for (which in case of a mutation leads to cystic fibrosis). In case of behavioural traits such as smoking/addiction and even more certain personality types, the relationship may be complex potentially implicating different genes and environmental factors. Therefore the relationship between genetics, smoking status and behavioural type has to be better explained: Is it an association or causality? How is the causality exactly? What can the methods used demonstrate: association or causality? How clear and strong is the relationship? What conclusions can be drawn from the study that are scientifically sound?

Further comments:

-          The HINT genes are also implicated in the pathogenesis of schizophrenia. Individuals with schizophrenia are often strong smokers. It may be interesting to conduct a study on such a subpopulation to investigate the genetics and relationship to addictive behaviour.

-          It would be good to introduce and explain scientific terms (such as “linkage disequilibrium” etc.) to increase readability and reach a larger reader audience.

-          2 Decimal places are normally enough for numbers.

-          L86-88: “Furthermore, chronic nicotine administration in mice increased HINT1 expression in the nucleus accumbens (NAc). However, this effect can be reversed by the administration of the nicotine antagonist, mecamylamine, after 24 hours or after 72 hours of nicotine withdrawal”. The causality should be better explained. The sentence suggests that Nicotine induces the gene expression but what does this proof exactly?

In conclusion, despite a strong academic presentation, the rationale and logic of the study has to be better explained. 

Comments on the Quality of English Language

English generally fine, minor grammar issues. 

Author Response

Dear Reviewer,

We would like to thank you for your valuable comments on the article. Below, you will find our reply to your review. All changes are with a description or a comment, changes have been made to the manuscript (track changes in the tracking group on the review tab).

Comments and Suggestions for Authors

The study “Exploring the Interplay of HINT1 Gene Polymorphisms, Smoking Behavior, and Personality Traits: A Haplotype Case-Control Study” was submitted to MDPI International Journal of Molecular Sciences.

The study is excellently written and demonstrates a methodological approach to study gene-behaviour interactions. Finding a genetic basis for smoking is relevant because, potentially, this may in the future lead into a gene therapy approach for individuals with a high dependence on nicotine or at least a better understanding of the underlying mechanisms. Understanding underlying mechanisms of addiction is therefore significant and relevant for the readers of Molecular Sciences.

On a methodological level, I would kindly like to ask for better explanation of the research questions, hypothesis and rationale. Specifically, the causal relationships between genetic, smoking status and personality trait are not clear to me. Causal relationships between a gene/allel and a certain trait are easier to demonstrate if they are very directly linked such as the CFTR gene and the transmembrane protein it codes for (which in case of a mutation leads to cystic fibrosis). In case of behavioural traits such as smoking/addiction and even more certain personality types, the relationship may be complex potentially implicating different genes and environmental factors. Therefore the relationship between genetics, smoking status and behavioural type has to be better explained: Is it an association or causality? How is the causality exactly? What can the methods used demonstrate: association or causality? How clear and strong is the relationship? What conclusions can be drawn from the study that are scientifically sound?

Thank you for these questions. The factors leading to the development of substance dependency are very complex and involve genetics, epigenetics and environment. The three main groups of factors could be divided further into thousands of genes and additional factors e.g. personality traits. In our study, we analysed personality (which is partially associated with genetics) and three single nucleotide polymorphic sites located in one gene in a group of smoking individuals and compared their results with never-smoking controls. In substance dependency phenotypes, the literature rarely states undoubted dichotomous causality due to the intricate interplay of hundreds of factors that together can contribute to the development of dependency. Hence, in the presented study, we do not use the term causality; instead, we analyse and describe the associations regarding alleles, genotypes, haplotypes, personality traits and their interactions. Regarding the clarity and strength of the associations – the power analysis and the p-value can give us some information, but we still are in the dark regarding the factual molecular mechanism and clinical relevance of the findings.  The HINT1 gene is not well known, and the literature regarding its involvement in neuropsychiatric conditions is not broad. We concluded our manuscript with adequate information regarding the revealed associations. The introduction, aim, and last paragraph of the discussion addressing limitations and conclusions have been rephrased for clarity and more scientific soundness – lines 81-120, 140-158, 164-169, 415-426, 488-504.

Further comments:

-          The HINT genes are also implicated in the pathogenesis of schizophrenia. Individuals with schizophrenia are often strong smokers. It may be interesting to conduct a study on such a subpopulation to investigate the genetics and relationship to addictive behaviour.

Thank you for this comment. The authors agree that a study regarding nicotine-dependent individuals with schizophrenia would be very interesting and would add to our understanding of the HINT1 gene's influence on both of these conditions.

-          It would be good to introduce and explain scientific terms (such as “linkage disequilibrium” etc.) to increase readability and reach a larger reader audience.

Thank you for this suggestion. The comment on linkage disequilibrium was added in lines 300-301.

-          2 Decimal places are normally enough for numbers.

Thank you for this suggestion. To maintain the accuracy of the presented data, the numbers were deducted to four decimal places.

-          L86-88: “Furthermore, chronic nicotine administration in mice increased HINT1 expression in the nucleus accumbens (NAc). However, this effect can be reversed by the administration of the nicotine antagonist, mecamylamine, after 24 hours or after 72 hours of nicotine withdrawal”. The causality should be better explained. The sentence suggests that Nicotine induces the gene expression but what does this proof exactly?

Thank you for this question. The paragraph in question refers to the work of Jackson et al. (2011) regarding the HINT1 gene polymorphic sites and HINT1 expression in the brain region associated with smoking. The results of this study indeed suggest that HINT1 gene expression changes after nicotine usage or administration. Unfortunately, the causality is not known, and even the Authors of the study do not explain it and conclude their study with information regarding the observed undoubted association and involvement of the HINT1 gene in the mechanisms of smoking. These results, taken together, through the presented associations, point us in the direction of gathering a solid justification for the analysis of HINT1 gene variants in subjects using nicotine.

In conclusion, despite a strong academic presentation, the rationale and logic of the study has to be better explained.

Thank you for this comment. The aim has been rephrased, and more information has been added in the introduction to support the rationale– lines 81-120, 164-169.

Comments on the Quality of English Language. English generally fine, minor grammar issues.

Thank you for pointing this out. The grammar has been corrected.

Reviewer 3 Report

Comments and Suggestions for Authors

The study presents potentially relevant data, but some aspects need major improvement.

1) The rationale for the study should be better articulated concerning a lack or presence of preexisting contradictory data.

2) Discussion of the possible causes of all the discrepancies between the results obtained in the present study and the preexisting literature is not sufficiently elaborated. The authors should try to find plausible explanations.

3) The sex factor never appears in the analyses except when a significant effect is to be emphasized (i.e., sex-stratified analysis). Likewise, other critical demographic variables, such as age or characterizing individual dependence, are not considered. For example, authors do not take age, onset age, intensity of addiction, or duration into account.

Reanalyze the data with these variables in mind.

4) I understand that this study examines genetic factors. However, there is still abundant literature that questions environmental factors, such as socioeconomic status, which are highly relevant for nicotine addiction, which the study does not consider at all. Therefore, researchers need to supplement the literature. If they can retrieve it post-hoc, they should include socioeconomic status among the variables considered or highlight its absence among the study's limitations.

Author Response

Dear Reviewer,

We would like to thank you for your valuable comments on the article. Below, you will find our reply to your review. All changes are with a description or a comment, changes have been made to the manuscript (track changes in the tracking group on the review tab).

Comments and Suggestions for Authors

The study presents potentially relevant data, but some aspects need major improvement.

1) The rationale for the study should be better articulated concerning a lack or presence of preexisting contradictory data.

Thank you for this comment. The aim has been rephrased, and more information has been added in the introduction to support the rationale– lines 81-120, 140-158, 164-169.

2) Discussion of the possible causes of all the discrepancies between the results obtained in the present study and the preexisting literature is not sufficiently elaborated. The authors should try to find plausible explanations.

Thank you for your insightful comments and for the opportunity to further discuss the findings of our study. Our research on the HINT1 gene three SNP haplotype analysis in nicotine-dependent and control individuals indeed yielded results that diverge from the existing literature. We acknowledge that the molecular mechanisms of the HINT1 gene are not yet fully understood. Recent research suggests that the HINT1 gene could be involved in various aspects of plasticity, mood regulation, anxiety-like behaviour, and stress-coping mechanisms. However, these mechanisms are complex and multifaceted, and our understanding of them is still evolving.

Several factors could explain the differences in our findings compared to previous studies. These may include differences in the study design, sample size, and population characteristics (such as age, gender, ethnicity, and environmental factors). We agree that it would be beneficial to discuss the potential reasons for these differences. However, without additional data or studies, any discussion would be largely speculative. We believe it is crucial to base our conclusions on solid evidence to maintain the integrity and reliability of our research.

Discussion regarding the sex-stratified analysis was added – lines 356-369.

3) The sex factor never appears in the analyses except when a significant effect is to be emphasized (i.e., sex-stratified analysis). Likewise, other critical demographic variables, such as age or characterizing individual dependence, are not considered. For example, authors do not take age, onset age, intensity of addiction, or duration into account.

Reanalyze the data with these variables in mind.

Thank you for this suggestion. Additional analysis was performed regarding the age, strength of dependency, and personality traits (Table 4). Unfortunately, we did not collect information on the age of onset, duration of nicotine usage, or socioeconomic information.

4) I understand that this study examines genetic factors. However, there is still abundant literature that questions environmental factors, such as socioeconomic status, which are highly relevant for nicotine addiction, which the study does not consider at all. Therefore, researchers need to supplement the literature. If they can retrieve it post-hoc, they should include socioeconomic status among the variables considered or highlight its absence among the study's limitations.

Thank you for this comment. The information regarding socioeconomic status was added to the introduction and study limitations (lines 59 and 424-426).

Round 2

Reviewer 2 Report

Comments and Suggestions for Authors

Thank you for adapting the manuscript as part of the first review round. In the second review round, the main topic should be to better define the contribution that the work makes to the scientific knowledge of the field. It has been explained by you as part of the responses that causal relationships and clinical relevance or significance cannot be determined and that the relationsships between genes and behavioural traits are complex. The manuscript mainly describes associations using statistical methods. 

Where do the authors see the problem that is being solved with the work? What is the exact hypothesis and can it be verified or falsified? What are the precise responses to the most central research questions on the topic? How can it be confirmed that the results are not a consequence of multiple statistical tests? In addition, I would like to ask for an argumentation why the research should be published? How does it advance the field (impact and relevance)? What research could be conducted later on based on the findings of this publication? How should a scientific proof look like for this topic? How does this paper provide a valid starting point for further research (purpose)?

In-text corrections: 

L 219 cont.: "We observed a significant statistical effect of the rs2551038 genotype interaction and cigarette-using status on the NEO-FFI Conscientiousness scale (F2.526 = 4.702; p = 0.0095; η2 = 0.018). The power observed for this factor was 79%, and approximately 2% was explained by the rs2551038 polymorphism and cigarette-using status on Conscientiousness trait score variance. The results are presented in Table 5."
The wording does not make it very clear if "genotype interaction" refers to a statistical term or if it describes a molecular mechanism? Please define the interaction precisely including relevance for the study. 

Comments on the Quality of English Language

English generally fine. 

Author Response

Dear Reviewer,

We would like to thank you for your valuable comments on the article. Below, you will find our reply to your review. All changes are with a description or a comment; changes have been made to the manuscript (track changes in the tracking group on the review tab).

Comments and Suggestions for Authors

Thank you for adapting the manuscript as part of the first review round. In the second review round, the main topic should be to better define the contribution that the work makes to the scientific knowledge of the field. It has been explained by you as part of the responses that causal relationships and clinical relevance or significance cannot be determined and that the relationsships between genes and behavioural traits are complex. The manuscript mainly describes associations using statistical methods.

Where do the authors see the problem that is being solved with the work?

Thank you for your question. The study aims to address the complex problem of understanding the interplay between genetic factors, personality traits, and cigarette smoking behaviour. By investigating the association between rs2551038 and smoking behaviour, the study aims to identify potential genetic markers that may influence an individual’s propensity to smoke. Examining the relationship between certain haplotypes and smoking could provide insights into the genetic complexity underlying smoking behaviours. Also, the presented research attempts to discern whether certain personality profiles are more common among smokers by comparing personality traits measured by the NEO Five-Factor Inventory in smokers and non-smokers. The study explores how genetic variants may interact with environmental factors (i.e. cigarette smoking) and personality traits, which helps understand the multifactorial nature of smoking. The study addresses whether genetic associations with smoking behaviour differ between males and females by conducting sex-stratified analyses. Overall, the problem being solved is to enhance the understanding of the biological and psychological factors that contribute to smoking behaviour.

What is the exact hypothesis and can it be verified or falsified?

Thank you for your question. The exact hypotheses for the presented study are as follows:

  1. Hypothesis 1: There is a significant association between the rs2551038 and cigarette smoking behaviour.
  2. Hypothesis 2: The haplotype consisting of rs3864236, rs2526303, and rs2551038 is significantly associated with cigarette smoking behaviour.
  3. Hypothesis 3: There are significant differences in personality traits, as measured by the NEO Five-Factor Inventory, between cigarette users and never-smokers.
  4. Hypothesis 4: There is a significant interaction effect between the genetic variants of rs2551038, personality traits, and cigarette smoking.

The hypotheses have been framed within the study's aims and tested through the research process: genotyping, psychometric testing, and statistical analysis of the obtained results. The hypotheses can be verified and falsified through the statistical analysis of obtained results.

What are the precise responses to the most central research questions on the topic?

Thank you for your question. The precise responses to the most central research questions are as follows:

  1. There is no association between rs2551038 genotypes or alleles and smoking in our study group;
  2. There is a significant difference in personality traits scores measured by the NEO-FFI in our study group: Extraversion was higher in cigarette users, Openness, Agreeability, and Conscientiousness were lower in cigarette users;
  3. There are weak but significant correlations of Fagestrom test results, and openness scale scores with age in our study group of cigarette users;
  4. Cigarette users with the C/C genotype obtained significantly lower scores on the Conscientiousness scale compared to the controls C/C carriers;
  5. The GTC haplotype was significantly more frequent in male never-smokers than in male cigarette users.

How can it be confirmed that the results are not a consequence of multiple statistical tests?

Thank you for your question. To ensure that the results are not a consequence of multiple statistical tests the Bonferroni correction has been applied to calculations where the multiple comparisons problem might have occurred – the NEO-FFI results analysis (Table 3), the Pearsons linear correlation (Table 4), the ANOVA (Table 5), the post hoc analysis (Table 6), LD (Table 7-9), the haplotype analysis (Table 10-12). The descriptions of the tables, relevant parts of the discussion and abstract reflect the amendments made.

In addition, I would like to ask for an argumentation why the research should be published?

Thank you for your question. We would like to highlight several key aspects that underscore the importance and novelty of our study:

  1. Our research provides a novel analysis of the interplay between HINT1 gene polymorphisms, smoking behaviour, and personality traits. This multidimensional approach offers a comprehensive understanding of the factors contributing to nicotine dependence.
  2. The study advances the field of genetic research by exploring the association between specific gene polymorphisms and cigarette smoking, which has not been extensively studied before. Our findings on the rs3864236 – rs2526303 – rs2551038 haplotype and its interaction with smoking behaviour add valuable insights to the genetic factors influencing nicotine dependency.
  3. Our research contributes to the psychological understanding of substance use by examining personality traits measured by the NEO Five-Factor Inventory. The significant differences in personality trait scores between cigarette users and never-smokers provide a deeper understanding of the psychological profiles associated with smoking.
  4. Our sex-stratified haplotype analysis reveals significant differences in haplotypes between sexes among smokers and never-smokers. This finding is particularly important as it suggests that genetic predisposition to nicotine dependence may vary by sex, which could lead to more personalized approaches to treatment. Additionally, this finding is in line with existing literature showing sex-specific associations of the HINT1 gene.
  5. The statistical analysis, including the observed significant linkage disequilibrium and the interaction effects on personality traits, demonstrates the methodological rigour of our study. These results provide a foundation for future research in this area.

In conclusion, our study offers a unique contribution to the fields of genetics and psychology by elucidating the complex relationships between genetic variants, smoking behaviour, and personality traits.  We believe that the scientific merit and relevance of our research make it a valuable addition to the literature and warrant its publication.

How does it advance the field (impact and relevance)?

Thank you for your question. Our research advances the field in several significant ways:

  1. By investigating the HINT1 gene polymorphisms and their association with smoking behaviour, our study provides new genetic insights that could be crucial for understanding the biological underpinnings of nicotine dependence. This can lead to the development of a dense set of genetic markers for identifying individuals at higher risk of developing nicotine addiction.
  2. The study’s findings on the correlation between personality traits and smoking behaviour contribute to the behavioural sciences by offering a more nuanced understanding of the psychological factors that may influence smoking habits. This knowledge is vital for creating more effective behavioural interventions.
  3. Our methodological approach, which includes a haplotype case-control study and a sex-stratified analysis demonstrates how complex interactions between genetics and behaviour can be analysed.
  4. The research has implications for public health, particularly in the realm of smoking cessation programs. By highlighting the role of personality traits, sex and genetic predispositions, our study suggests that personalized approaches to smoking cessation could be more effective.
  5. Our work exemplifies the benefits of interdisciplinary research, combining genetics, psychology, and public health. This collaborative approach can inspire similar studies across disciplines, leading to a more integrated understanding of health and behaviour.

In summary, our study not only fills a gap in the current literature but also provides a foundation for future research, public health initiatives, and interdisciplinary collaboration.

What research could be conducted later on based on the findings of this publication?

Thank you for your question. Future research could take several directions to further explore and validate the interplay between genetics, smoking behaviour, and personality traits. Here are some potential avenues for subsequent studies:

  1. Conducting longitudinal research to track changes in smoking behaviour over time in individuals with different HINT1 gene polymorphic variants. This could provide insights into the long-term effects of these genetic variants.
  2. Exploring additional polymorphic sites in the HINT1 gene or neighbouring genes that may contribute to nicotine dependence. A broader genetic analysis could uncover more complex genetic networks involved in smoking behaviour. Additionally, the analysis of HINT1 promoter methylation could be insightful in terms of understanding its function in substance-dependent phenotypes.
  3. Investigating the biological mechanisms by which the HINT1 gene polymorphisms influence smoking behaviour. Functional studies could include cellular or animal models to understand the gene’s role in nicotine addiction.
  4. Examining the associations found in our study across different cultures and populations could help determine whether the observed relationships are consistent across diverse groups or if cultural factors play a significant role.
  5. Testing personalized intervention strategies based on the personality profiles identified in our study. This could involve developing tailored smoking cessation programs and evaluating their effectiveness.
  6. Researching how individuals with different HINT1 gene variants respond to various pharmacological treatments for smoking cessation. This could lead to more personalized medicine approaches.
  7. Utilizing neuroimaging techniques to observe brain activity patterns associated with the HINT1 gene polymorphic variants in smokers and non-smokers. This could reveal neural correlates of the genetic predisposition to smoking.
  8. Performing meta-analyses that combine data from multiple studies to achieve a higher statistical power and validate the findings of our study on a larger scale.
  9. Investigating how the findings of our study can inform public health policies and smoking prevention efforts. This could involve collaboration with policymakers to translate research into practice.

Information added in lines 420 – 441.

How should a scientific proof look like for this topic?

Thank you for your question. For the topic of genetic and personality factors in cigarette smoking behaviour, scientific proof should consist of several elements. The study should have a well-defined design, optimized methodology allowing for replication, thorough statistical analyses, be consistent with existing literature, and have biological and psychological plausibility. In this context, scientific proof is not just a single study but a body of evidence that collectively supports the conclusions drawn. It involves validation and verification through multiple studies and analyses, contributing to a comprehensive understanding of the topic. Hence, the presented study adds to our understanding of the genetics and psychology of nicotine-using.

How does this paper provide a valid starting point for further research (purpose)?

Thank you for your question. Our study serves as a starting point for future investigations for several reasons:

  1. Our research has revealed a specific HINT1 gene haplotype that is associated with smoking behaviour. This provides a molecular target for subsequent studies aiming to understand the genetic basis of nicotine dependence.
  2. The interactions we found between the HINT1 genetic variant and personality trait offer a new perspective on the behavioural aspects of smoking. Future research can build on this by exploring these links in different populations or under various environmental conditions.
  3. The methodologies we employed can serve as a blueprint for future studies. Researchers can replicate our approach to validate our findings or apply it to different genetic and behavioural studies.
  4. The data generated from our study can be included in larger meta-analyses to confirm the associations we observed. This could help establish more definitive conclusions about the role of the HINT1 genetic variants in smoking behaviour.
  5. Our study's interdisciplinary nature, which spans genetics, psychology, and public health, provides a model for cross-disciplinary research. It encourages collaboration between fields to tackle complex health issues like smoking.

In essence, our paper lays the groundwork for a wide array of research possibilities, from genetic studies to public health interventions. It offers a comprehensive and methodologically sound basis for further exploration into the intricate relationship between genetics, personality, and behaviour.

In-text corrections:

L 219 cont.: "We observed a significant statistical effect of the rs2551038 genotype interaction and cigarette-using status on the NEO-FFI Conscientiousness scale (F2.526 = 4.702; p = 0.0095; η2 = 0.018). The power observed for this factor was 79%, and approximately 2% was explained by the rs2551038 polymorphism and cigarette-using status on Conscientiousness trait score variance. The results are presented in Table 5."

The wording does not make it very clear if "genotype interaction" refers to a statistical term or if it describes a molecular mechanism? Please define the interaction precisely including relevance for the study.

Thank you for your question.  We would like to clarify that in the context of our study, “genotype interaction” refers to a statistical interaction term used in our analysis. Specifically, it denotes the combined effect of the rs2551038 polymorphism and cigarette-using status on the NEO-FFI Conscientiousness scale scores.

To elaborate, the interaction term in our statistical model assesses whether the effect of one variable (in this case, the rs2551038 polymorphism) on the outcome (Conscientiousness trait score) changes depending on the level of another variable (cigarette-using status). This is a standard approach in statistical analysis to explore if the relationship between a predictor and an outcome is different across groups defined by another variable.

The significance of this interaction (F2,526 = 4.702; p = 0.0095; η^2 = 0.018) suggests that the impact of the rs2551038 polymorphism on Conscientiousness is not uniform across all individuals but varies depending on whether they are cigarette users or non-users. The observed power of 79% indicates that our study had a high probability of detecting an effect of this size, should it exist. Furthermore, the eta-squared (η^2) value of 0.018 implies that approximately 2% of the variance in Conscientiousness trait scores can be explained by the interaction between the rs2551038 polymorphism and cigarette-using status.

This finding is relevant to our study as it provides evidence that genetic factors may influence personality traits differently in individuals who smoke compared to those who do not. It highlights the importance of considering the context of behavioural factors when examining the effects of genetic polymorphisms.

Comments on the Quality of English Language

English generally fine.

Thank you for your comment.  Minor mistakes were corrected.

Reviewer 3 Report

Comments and Suggestions for Authors

The authors have appropriately revised the manuscript, which is now acceptable.

Author Response

Dear Reviewer,

We would like to thank you for your valuable comments on the article. Below, you will find our reply to your review.

Comments and Suggestions for Authors

The authors have appropriately revised the manuscript, which is now acceptable.

The authors would like to thank the Reviewer for their contribution to the improvement of the manuscript.

Round 3

Reviewer 2 Report

Comments and Suggestions for Authors

Thank you for providing more information on the hypothesis and central research questions. This is helpful to obtain a better understanding of the research. I must note, however, that all these hypotheses and research questions have been explicitly or implicitly answered by large genome-wide association studies (GWAS). The authors have conducted a candidate-gene study and used a behavioural assessment (psychometric tests) to study the link between genetics and phenotypical presentation. The rationale for selecting the specific gene variants in the present study is not elucidated – this would be very important to know.

The link between genetics and smoking is complex with more than 2500 genetic variants linked to smoking. It is well known that smoking and certain personality types/traits are associated.

Essentially, I have the impression that there is an unnecessary triangulation in the present study due to false definition of hypothesis.

There is most likely a complex causal relationship between genetics and smoking. (People with a certain genetic configuration are more likely to smoke).

There is most likely a complex causal relationship between personality types and smoking. (People with a certain personality type are more likely to smoke).

What is not clear in the present study is why it was decided to further triangulate the relationship (Genes – Behaviour – Smoking status).

Specifically, there should be one study which examines what genes lead to smoking.

A second study could be conducted to understand which personality type is linked, with an increased likelihood, to smoking.

The question of whether a specific gene leads, at the same time to “smoking” and a “high risk for a smoking personality type” does not respond to the question of whether “smoking” in a given person is caused by genetics or the behavioural type. I assume that this was the question of interest which may have led to this triangulation.

It may be interesting to find out if a person with a specific genetic has a very high risk for smoking (preponderance for genetic cause as opposed to behavioural cause). In this case,  I would expect very strong statistical associations in GWAS between a specific genetic marker and smoking status. If this strong association had been found, it could be interesting to further study the molecular pathways to identify a target for treatment.

If the triangulation that is described is a component of the research article, it would be better to take steps to resolve this triangulation if possible. Otherwise, the article could lead to unnecessary confusion in the readers. Please let me know as part of the next revision round how the triangulation could be resolved and how you would like to proceed further with the article. 

Comments on the Quality of English Language

English is fine, minor issues

Author Response

ANSWER

Dear Reviewer,

We would like to thank you for your valuable comments on the article. Below, you will find our reply to your review. All changes are with a description or a comment; changes have been made to the manuscript (track changes in the tracking group on the review tab).

Comments and Suggestions for Authors

Thank you for providing more information on the hypothesis and central research questions. This is helpful to obtain a better understanding of the research. I must note, however, that all these hypotheses and research questions have been explicitly or implicitly answered by large genome-wide association studies (GWAS). The authors have conducted a candidate-gene study and used a behavioural assessment (psychometric tests) to study the link between genetics and phenotypical presentation. The rationale for selecting the specific gene variants in the present study is not elucidated – this would be very important to know.

The link between genetics and smoking is complex with more than 2500 genetic variants linked to smoking. It is well known that smoking and certain personality types/traits are associated.

Essentially, I have the impression that there is an unnecessary triangulation in the present study due to false definition of hypothesis.

There is most likely a complex causal relationship between genetics and smoking. (People with a certain genetic configuration are more likely to smoke).

There is most likely a complex causal relationship between personality types and smoking. (People with a certain personality type are more likely to smoke).

What is not clear in the present study is why it was decided to further triangulate the relationship (Genes – Behaviour – Smoking status).

Specifically, there should be one study which examines what genes lead to smoking.

A second study could be conducted to understand which personality type is linked, with an increased likelihood, to smoking.

The question of whether a specific gene leads, at the same time to “smoking” and a “high risk for a smoking personality type” does not respond to the question of whether “smoking” in a given person is caused by genetics or the behavioural type. I assume that this was the question of interest which may have led to this triangulation.

It may be interesting to find out if a person with a specific genetic has a very high risk for smoking (preponderance for genetic cause as opposed to behavioural cause). In this case,  I would expect very strong statistical associations in GWAS between a specific genetic marker and smoking status. If this strong association had been found, it could be interesting to further study the molecular pathways to identify a target for treatment.

 If the triangulation that is described is a component of the research article, it would be better to take steps to resolve this triangulation if possible. Otherwise, the article could lead to unnecessary confusion in the readers. Please let me know as part of the next revision round how the triangulation could be resolved and how you would like to proceed further with the article. 

Comments on the Quality of English Language

English is fine, minor issues

Dear Reviewer,

Thank you very much for another round of comments, as it turned out to be valuable for our study. The scientific team has thought about and discussed the last comments very carefully, and after discussion, we have come to the conclusion that they are valid and justified. In fact, it would be better to show the association of genetics with smoking without including personality traits in the analysis. Both phenotypic traits are multigene and multifactorial, so the analysis may be more complicated. Statistical calculations and results shown were changed according to the Reviewer's recommendations (Results section, Materials and Methods section 4.4).

Regarding the selection of the gene in our study, this is justified in the Manuscript (line 81-132). However, we would add that the selection of polymorphic variants of the HINT1 gene was deeply considered. The first author is interested in this gene locus in the context of nicotine addiction. A scientific project was submitted for the present study, which received high scores and funding in a national competition to study this particular gene. It is in the field of special interest and research field of lead author Dr Suchanecka.

Of course, we are also interested in other genes and their polymorphic variants, but HINT1 is still poorly described in research papers, so we pay special attention to it.

Once again, we thank you for these comments, we agree with their validity and send the Manuscript with corrections.